# Cross-Talk between the (Endo)Cannabinoid and Renin-Angiotensin Systems: Basic Evidence and Potential Therapeutic Significance

**DOI:** 10.3390/ijms23116350

**Published:** 2022-06-06

**Authors:** Krzysztof Mińczuk, Marta Baranowska-Kuczko, Anna Krzyżewska, Eberhard Schlicker, Barbara Malinowska

**Affiliations:** 1Department of Experimental Physiology and Pathophysiology, Medical University of Białystok, ul. Mickiewicza 2A, 15-222 Białystok, Poland; krzysztof.minczuk@umb.edu.pl (K.M.); mabar@umb.edu.pl (M.B.-K.); anna.krzyzewska@umb.edu.pl (A.K.); 2Department of Pharmacology and Toxicology, University of Bonn, Venusberg-Campus 1, 53127 Bonn, Germany

**Keywords:** cannabinoids, endocannabinoids, angiotensin II, angiotensin 1-7, RAS, COVID-19

## Abstract

This review is dedicated to the cross-talk between the (endo)cannabinoid and renin angiotensin systems (RAS). Activation of AT_1_ receptors (AT_1_Rs) by angiotensin II (Ang II) can release endocannabinoids that, by acting at cannabinoid CB_1_ receptors (CB_1_Rs), modify the response to AT_1_R stimulation. CB_1_R blockade may enhance AT_1_R-mediated responses (mainly vasoconstrictor effects) or reduce them (mainly central nervous system-mediated effects). The final effects depend on whether stimulation of CB_1_Rs and AT_1_Rs induces opposite or the same effects. Second, CB_1_R blockade may diminish AT_1_R levels. Third, phytocannabinoids modulate angiotensin-converting enzyme-2. Additional studies are required to clarify (1) the existence of a cross-talk between the protective axis of the RAS (Ang II—AT_2_ receptor system or angiotensin 1-7—Mas receptor system) with components of the endocannabinoid system, (2) the influence of Ang II on constituents of the endocannabinoid system and (3) the (patho)physiological significance of AT_1_R-CB_1_R heteromerization. As a therapeutic consequence, CB_1_R antagonists may influence effects elicited by the activation or blockade of the RAS; phytocannabinoids may be useful as adjuvant therapy against COVID-19; single drugs acting on the (endo)cannabinoid system (cannabidiol) and the RAS (telmisartan) may show pharmacokinetic interactions since they are substrates of the same metabolizing enzyme of the transport mechanism.

## 1. Introduction

In recent years more and more publications have appeared regarding the cross-talk between the (endo)cannabinoid and renin angiotensin (RAS) systems. This issue has become even more important in the era of the coronavirus disease 2019 (COVID-19) pandemic since angiotensin-converting enzyme type 2 (ACE2), one of the key RAS components, was described as a receptor required for the entry of the severe acute respiratory syndrome coronavirus 2 (SARS-CoV-2) into host cells [1,2,3,4]. The primary aim of our review is to compare all publications regarding the interaction between (endo)cannabinoids and RAS and to find as many common effects and mechanisms as possible. The secondary aim is dedicated to the determination of its potential therapeutic significance.

## 2. Materials and Methods

To find the most relevant articles dealing with the interaction between angiotensin and cannabinoid systems, we performed a comprehensive search in the PubMed database (closed on 20 May 2022). The following key phrases were used in the search engine: “angiotensin cannabinoid” (which yielded 94 results), “angiotensin 1-7 cannabinoid” (4 results), “ACE cannabinoid” (15 results), “ACE2 cannabinoid” (13 results), “ACE2 cannabinoid COVID” (11 results) and “cannabinoid SARS-CoV-2” (68 results). Taking approved therapeutic usage into consideration, search phrases also consisted of: “sativex”, “nabilone”, “dronabinol”, and “epidiolex” coupled with the word “angiotensin” (a total of 13 results).

First, the titles and then the abstracts and full texts of the identified papers were analyzed, and duplicated articles or those with non-relevant content were removed from the list. Only articles in English were considered. Non-relevant studies included, for example, no described interaction between the two systems of interest or “ACE” being an acronym for unrelated definitions such as “central amygdala”. Finally, 43 publications were included in this review, which can be seen systematically summarized in Table 1, Table 2, Table 3 and Table 4. Additional papers listed in the References were used to provide more background on angiotensin and cannabinoid systems, as well as SARS-CoV-2. The screening process is summarized in Figure 1, illustrating the PRISMA flowchart [5].

To make reading fluent, the current article is structured into paragraphs, first describing the cannabinoid and angiotensin systems (Section 3.1.1 and Section 3.1.2) and then the interaction between both systems (Section 3.2, Section 3.3, Section 3.4, Section 3.5). Interactions are being described in the following order: overall cross-talk between RAS and eCBS (Section 3.2 and Table 2), the interaction between angiotensin 1-7 and eCB-mediated effects (Section 3.3 and Table 3), and interaction relevant to COVID-19 (Section 3.4 and Table 4).

## 3. Results and Discussion

### 3.1. Components and Main Effects of (Endo)Cannabinoid and Renin-Angiotensin Systems

#### 3.1.1. (Endo) Cannabinoid System

Cannabinoids are lipophilic compounds that belong to three groups: (1) phytocannabinoids, i.e., compounds naturally occurring in the hemp plant Cannabis sativa var. indica, including the psychoactive Δ^9^-tetrahydrocannabinol (THC) and non-intoxicating compounds such as cannabidiol (CBD), cannabigerol (CBG) or cannabinol (CBN); (2) synthetic compounds, e.g., CP55940, WIN55212-2 and JWH133 and (3) endocannabinoids (eCBs), e.g., anandamide (AEA) or 2-arachidonoylglycerol (2-AG), or endocannabinoid-like compounds, e.g., palmitoylethanolamide (PEA). eCBs and endocannabinoid-like compounds are produced by the body; eCBs have an affinity to cannabinoid receptors (CBRs), whereas endocannabinoid-like compounds, despite their similar chemical structure to eCBs, do not have an affinity for CBRs [6,7,8].

Cannabinoids exert their effects mainly via two G protein-coupled receptors (GPCRs): cannabinoid CB_1_ (CB_1_Rs) and CB_2_ (CB_2_Rs) receptors. Both CB_1_ and CB_2_ receptors act via G_i/o_-protein-dependent pathways and inhibition of adenylyl cyclase. Moreover, their activation may lead to stimulation of mitogen-activated protein kinases (MAPK) and, in the case of CB_1_Rs, modulation of calcium and potassium channels [6,9,10]. The most abundant cannabinoid receptors are CB_1_Rs, which are expressed mainly in the central nervous system (CNS) and also in low but functional levels in most peripheral tissues, including peripheral neurons, the cardiopulmonary system (systemic and pulmonary vessels), kidney, liver, and other tissues. CB_2_Rs are localized mainly peripherally, especially on immune cells. They inhibit inflammation and oxidative stress [6,7,9,11,12,13]. Cannabinoids also act via other types of G protein-coupled receptors (GPR55 and GPR18) and other receptor families (ionotropic vanilloid TRPV1 and peroxisome proliferator-activated receptor (PPAR) [6,9,10].

Cannabinoid receptors and eCBs belong to the endocannabinoid system (Figure 2), including enzymes involved in the synthesis and degradation of eCBs described in detail in our previous review [8]. Because of the complexity of the synthesis and metabolism of eCBs, only those receptors and enzymes that are important for the understanding of the current review are described below and presented in Figure 2. Briefly, 2-AG is formed almost exclusively by hydrolysis of diacylglycerols (DAGs) by diacylglycerol lipase (DAGL). Cannabinoids exert a multidirectional effect on the body not only through interaction with appropriate receptors but also indirectly through metabolites resulting from their degradation. Fatty acid amide hydrolase (FAAH) is responsible for the degradation of AEA, while monoacylglycerol lipase (MAGL) is particularly involved in the degradation of 2-AG. As a result of their degradation, arachidonic acid (AA) may be formed and converted into eicosanoid metabolites by cyclooxygenase 1/2 (COX-1/COX-2) with a broad spectrum of activity in the body. The eCB system, with its widespread distribution in the body, represents an important signaling pathway involved in many biological processes. Moreover, the eCB system seems to be a promising point for new therapeutic strategies, in many cases, including FAAH and MAGL inhibitors such as URB597 and JZL184, respectively (for review, see [8]).

CB_1_R function undergoes modification under many pathological circumstances. Intriguingly, as shown in Figure 2, both positive and negative changes in their effects may occur under the same pathological condition (described in detail previously by [6,9,11,12,13,14]). Thus, CB_1_R activation may have beneficial effects against loss of appetite and body weight, nausea, spasticity in multiple sclerosis, pain (especially peripherally-restricted CB_1_R agonists), anxiety- and depressive-like behavior, posttraumatic stress disorder, neuroprotection, and epilepsy [9,15,16,17,18] or may lead to vasodilatation in arteries of hypertensive animals [19,20]. THC itself (international non-proprietary name, dronabinol), its synthetic analog nabilone, and the mixture of THC and cannabidiol (nabiximols) are approved by the US Food and Drug Administration and other national or supranational drug agencies for some of the above indications (Table 1 [21]). However, CB_1_R activation can also lead to cognitive impairment, agitation and acute psychosis [22,23], vasodilatation and/or hypotension in various forms of shock, decreased cardiac function (cardiomyopathies, and heart failure), diet-induced obesity, development of non-alcoholic fatty liver disease and peripheral insulin resistance [6,7,9,15]. Deleterious consequences of the activation CB_1_Rs result from their ability to increase reactive oxygen species (ROS) generation and pro-inflammatory responses leading to endothelial and cardiomyocyte cell remodeling/fibrosis, death, and cardiovascular, metabolic, renal, respiratory, and hepatic dysfunction, detected in both preclinical and clinical studies [6,9,11,13,15,24]. In addition to the agonists mentioned above, one CB_1_R antagonist, rimonabant, was available until 2008 as an anti-obesity drug (Table 1).

**Table 1 ijms-23-06350-t001:** Approved drugs targeting the (endo)cannabinoid and renin-angiotensin systems ^1^.

System	Class	Mechanism	Approved Drug	Indications
(endo)cannabinoid system	agonists	unselective cannabinoid receptor agonist	dronabinol (Δ^9^-tetrahydrocannabinol, THC)	anorexia and weight loss in HIV patients, nausea and vomiting in cancer chemotherapy
		unselective cannabinoid receptor agonist	nabilone	like dronabinol
	antagonists	CB_1_ receptor antagonist	rimonabant ^2^	obesity
	other	weak activity towards the cannabinoid system, antioxidant drug	cannabidiol	neuropathic pain; Lennox-Gastaut and Dravet syndrome
		(see dronabinol and cannabidiol)	nabiximols (1:1 formulation of dronabinol and cannabidiol)	neuropathic pain in multiple sclerosis, intractable cancer pain
renin-angiotensin system	agonists	unselective AT receptor agonist	angiotensin II	increase in blood pressure in adults with septic or other distributive shock
	antagonists	renin inhibitor	e.g., aliskiren	essential hypertension
		AT_1_ receptor antagonist	e.g., candesartan, valsartan	essential hypertension, congestive heart failure
		aldosterone receptor antagonist	e.g., eplerenone	congestive heart failure
	other	angiotensin converting enzyme inhibitor	e.g., perindopril	essential hypertension, congestive heart failure

^1^ Based essentially on references from Section 3.1.1, Section 3.1.2 and [21]. ^2^ Withdrawn from the market in 2008.

#### 3.1.2. Renin-Angiotensin System

The RAS plays a significant role in the regulation of blood pressure and sodium and water homeostasis and occurs as (1) circulating hormonal system, (2) local or paracrine RAS expressed in many organs (both in the CNS and in the periphery), and (3) intracellular or intracrine RAS in several cell types [25,26]. The RAS comprises different peptides with opposing biological effects (Figure 3). Briefly, angiotensinogen is converted into angiotensin I (Ang I) by renin and subsequently into angiotensin II (Ang II) by the angiotensin-converting enzyme (ACE). Ubiquitous actions of the main effector of the RAS, namely Ang II, are related to the activation of several signal transduction pathways, which are predominantly attributed to the Ang II type 1 receptor (AT_1_R). In addition, Ang II also acts through the Ang II type 2 receptor (AT_2_R), or is degraded by ACE2 to Ang 1-7 acting via the Mas receptor (MasR) [27,28]. All receptors are GPCRs; for details of the most important signaling pathways of AT_1_Rs, see Section 3.2.1.

As shown in Figure 3, the RAS consists of two counter-regulatory arms: the classic one (red rectangle; deleterious axis) containing ACE/Ang II/AT_1_Rs and the alternative one (green rectangles; beneficial axis) constituted by (i) AT_2_Rs and (ii) ACE2/angiotensin 1-7 and its MasRs. The effects following activation of the particular receptors are shown in Figure 3. The increase in the activity of one axis results in the decrease of the other [26,29,30]. Importantly, disturbance of the balance between the deleterious and the protective axis with the predominance of the ACE/Ang II/AT_1_R axis leads to many pathologies, including metabolic (diabetes mellitus), renal, cardiovascular (heart failure, myocardial infarction, hypertension), lung, hepatic, digestive, endocrine, neurodegenerative, hematological, reproductive, and muscular disease [4,26,30,31,32]. Even if the classic arm plays an important and unpleasant role in many disease states of humans, it is not per se deleterious. One should consider that well-being and even survival would not be possible if this arm were missing in the presence of severe loss of fluid and sodium and/or marked hypotension.

The RAS is the target of several (classes of) drugs (Figure 3, Table 1). Angiotensin II itself is used for the treatment of septic shock. Antagonists at the different levels of the RAS system, i.e., renin, angiotensin II and aldosterone, are used to treat essential hypertension and/or congestive heart failure. For the latter two indications, inhibitors of the degradation of angiotensin I to angiotensin II (ACE inhibitors) are important drugs.

### 3.2. Examples of Cross-Talk between the (Endo) Cannabinoid and Renin-Angiotensin Systems

Details of particular publications regarding the cross-talk between the (endo)cannabinoid and renin-angiotensin systems are shown in Table 2. We have concentrated mainly on changes in components of both systems but not on physiological or pathophysiological effects induced by activation of the individual receptors. For a description of all compounds mentioned in our review, see Figure 2 and Figure 3.

**Table 2 ijms-23-06350-t002:** Examples of the cross-talk between the (endo)cannabinoid and renin-angiotensin systems.

Species	Model	Agonist Concentration (μM) or Dose	Effect	(Functional) Antagonist; Concentration In Vitro (μM) or Dose	Influence on the Agonist Effect	Final Conclusion of the Authors	References
**cells: Chinese hamster; human; African green monkey**	CHO; HEK293; COS7 cells (co-expressing AT_1_Rs and CB_1_Rs) from ovaries, kidneys, and fibroblasts, respectively	Ang II (0.1)	↑2-AG ↔AEA ↑G_o_ protein activation	AM251 (10) THL (1)	↓Ang II-induced G_o_ protein activation	**AT_1_R stimulation leads to DAGL**-mediated transactivation of **CB_1_Rs** in an autocrine and paracrine manner	[33,34]
**cells** **: mouse**	neuro2A cells, a neuroblastoma cell line co-expressing CB_1_Rs and AT_1_Rs	Ang II (0.01–10)	↑pERK levels via G_αi_ instead of G_αq_ the expression of AT_1_R shifts CB_1_Rs from an intracellular compartment to the plasma membrane	losartan CB_1_R-targeting siRNA RIM (1) THL (1) HU210 (0.0001)	Ang II-induced ↑pERK ↓ by losartan, CB_1_R-targeting siRNA, RIM, and THL; ↑ by HU210 (occurring in the presence of a very low non-signaling concentration of Ang II only)	**AT_1_Rs and CB_1_Rs form receptor heteromers**; blocking CB_1_R activity prevented the Ang II-mediated pathologic effect	[35]
**cells** **: rats**	hepatic stellate cells from control rats (cHSCs) and rats treated with ethanol for 8 months (eHSCs)	Ang II (1)	CB_1_R, AT_1_R and AT_1_R-CB_1_ heteromer levels in eHSCs > cHSCs; ↑pERK levels, ↑mitogenic and ↑profibrogenic markers in eHSCs > cHSCs	RIM (1)	↓Ang II-induced changes		
**Blood Vessels**							
**rats** **Wistar**	aortic VSMCs	Ang II (0.1)	↑2-AG level ↑Ca^2+^ signal	THL (1) JZL184 (1)	↓ and ↑ of Ang II-induced 2-AG formation and Ca^2+^ signal by THL and JZL184, respectively	Ang II stimulates **eCB (2-AG) release** from the vascular wall that reduces the vasoconstrictor effects of Ang II via **CB_1_R activation** (eCBs act **as protective negative feedback** in response to Ang II)	[36]
**rats and/or** **mice**	aortic rings from rats aortic rings from CB_1_^−/−^ and WT mice	Ang II (0.001–0.1)	concentration-dependent contraction	WIN-2 (10) O2050 (1) THL (1) JZL184 (1)	vasodilation to WIN-2; not detected in CB_1_^−/−^ O2050, THL↑ , and JZL184↓ vasoconstrictor effect of Ang II; amplificatory effect of O2050 in WT only		
**rats and/or** **mice**	skeletal muscle arterioles, saphenous arteries	Ang II (0.001–0.1)	concentration-dependent contraction	WIN-2 (1) O2050 (1) RIM (1) AM251 (1) THL (1)	vasodilation to WIN-2; not detected in CB_1_^−/−^ ↑vasoconstrictor effect of Ang II in WT but not in CB_1_^−/−^	Ang II stimulates **eCB release** from the vascular wall that reduces the vasoconstrictor effects of Ang II via **CB_1_R activation** (eCBs act as **protective negative feedback** in response to Ang II)	[37]
**rats** **Wistar**	intramural coronary resistance arterioles	Ang II (0.0001–10)	concentration-dependent contraction	WIN-2 (0.0001–1) O2050 (1) THL (1)	vasodilatation to WIN-2 reduced by O2050 and AM251 ↑vasoconstrictor effect of Ang II	Ang II stimulates **eCB release** from the vascular wall that reduces the vasoconstrictor effects of Ang II via **CB_1_R activation** (eCBs act **as protective negative feedback** in response to Ang II)	[38]
**rats** **Wistar**	pulmonary arteries	Ang II (0.0001–0.03)	concentration-dependent contraction	AM251 (1) RHC80267 (40) JZL184 (1) URB597 (1)	AM251 and RHC80267 ↑ but JZL184 ↓ vasoconstrictor effect of Ang II; URB597 ↔	Ang II stimulates **eCB (2-AG) release** from the vascular wall that reduces the vasoconstrictor effects of Ang II via **CB_1_R activation** (eCBs act as **protective negative feedback** in response to Ang II)	[39]
**rats**	uterine artery from hypertensive TgA and normotensive SD rats	Ang II (0.00001–0.01)	concentration-dependent contraction, stronger in TgA	URB597 (1) JZL184 (1) RIM (1)	↓responses to Ang II in SD and TgA ↓responses to Ang II in TgA ↔responses to Ang II in SD and TgA	eCBs reduce the Ang II-induced contraction **in a CB_1_R-independent manner** in the early stages of hypertensive pregnancy (eCBs act as **protective negative feedback** in response to Ang II)	[40]
**rats SD** **mice**	VSMCs from rat and mouse thoracic aortas with CB_1_R expression	Ang II (1)	↑ROS production ↑NADPH oxidase activity	RIM (0.1–1) or AM251 (1) CP55940 (1)	↓AT_1_Rs and decrease in the Ang II-induced ↑ROS production and ↑NADPH oxidase activity ↑AT_1_Rs	CB_1_R inhibition (in vitro and in vivo) has atheroprotective effects by down-**regulation of AT_1_Rs**, decreased vascular ROS, and thus improved endothelial function in hypercholesterolemic ApoE^−/−^ mice	[41]
**mice**	ApoE^−/−^ treated with a cholesterol-rich diet		development of atherosclerotic plaques, ↓aorta relaxation, ↔aortic AT_1_R level	RIM (10 mg/kg/day; p.o.) for 7 weeks	↓aortic AT_1_Rs and improvement of endothelial function, no effect on atherosclerotic plaques		
**Heart**							
**rats SD**	isolated Langendorff-perfused hearts	Ang II (0.001–0.1) 2-AG (1) WIN-2 (1)	↓CF and moderate negative inotropic effect 2-AG and WIN-2: ↑CF WIN-2: negative inotropic effect	O2050 (1) + Ang II orlistat (10) + Ang II	↓cardiac effects of Ang II	besides direct cardiac responses, Ang II induces indirect ones via **eCBs** (probably 2-AG) activating **CB_1_Rs**: - direct positive inotropy reversed into a negative one, ↓oxygen demand - direct coronary constriction attenuated, (↑)oxygen supply (eCBs act as **protective negative feedback** in response to Ang II)	[42]
**mice**	streptozotocin-induced diabetes	diabetic cardiomyopathy	↑myocardial CB_1_ and AT_1_R expression and AEA level connected with cardiac dysfunction, inflammation, oxidative/nitrative stress	RIM or AM281 (10 mg/kg; i.p. daily for 11 weeks) or CB_1_R deletion (CB_1_^−/−^ mice)	pharmacological inhibition or genetic deletion of CB_1_Rs—improvement of diabetic cardiac dysfunction connected with ↓AT_1_Rs and CB_1_Rs in LV	overactivation of the eCB system and CB_1_Rs may play an important role in the pathogenesis of diabetic cardiomyopathy by **facilitating AT_1_R** expression and signaling	[43]
**rats** **Wistar**	isolated Langendorff-perfused hearts underwent ischemia + reperfusion	ischemia and reperfusion	↑stroke size, ↓ventricular function; ↑ cardiac AT_1_R level and ↔ cardiac AT_2_R level	CBD (5 mg/kg; i.p. daily for 10 days)	↓stroke size and ↑ventricular function; ↓ cardiac AT_1_R level and ↑ cardiac AT_2_R level	cardioprotective effect of CBD might result from an **increase in cardioprotective AT_2_Rs** stimulating counter-regulatory effects on the AT_1_Rs	[44]
**mice**	Ang II-induced fibrosis and inflammation	Ang II infusion (1 µg/kg/min [preventive] or 500 [therapeutic] for 4 weeks)	fibrosis and inflammation in the heart, aorta, lung, kidney, and skin	EHP-101 (2, 5 or 20 mg/kg for 4 or 2 weeks)	↓cardiac, aortic, lung, kidney, and skin fibrosis and inflammation in the preventive or therapeutic model	EHP-101 (dual agonist of CB_2_Rs and PPARγ) can alleviate cardiac, aortic, lung, kidney, and skin inflammation induced by Ang II	[45]
**Blood Pressure**							
**mice** **CB_1_^−/−^** **CB_1_^+/+^**	anesthetized	Ang II (1 μg/kg/min)	↑BP in WT and CB_1_^−/−^	O2050 (10 mg/kg; p.o.)	↑ pressor effect of Ang II in WT, not in CB_1_^−/−^	confirmation of in vitro experiments on isolated arteries that Ang II stimulates **release of eCBs (2-AG)** from the vascular wall that reduce vasoconstrictor effects of Ang II via **CB_1_R activation**	[36]
**rats SD**	conscious	Ang II-induced hypertension (60 ng/min; s.c. for 10–12 days)	↑BP	AM251 (3 mg/kg; i.v.) URB597 (10 mg/kg; i.v.) in pentobarbital- anaesthetized rats	AM251 ↑BP and URB597 ↓BP in Ang II-induced hypertension but not in normotension	the Ang II-induced hypertension is diminished by **eCBs acting at CB_1_Rs**; effect of URB597 reduced by AM251	[19]
**rats** **Wistar**	conscious	Ang II (500 ng/kg/h) + VP (50 ng/kg/h for 4 days)- induced hypertension	↑BP	AEA (3 mg/kg) URB597 (3 mg/kg) WIN-2 (150 μg/kg) AM251 (3 mg/kg)	AEA, WIN-2 ↓BP in Ang II-VP-induced hypertension; URB597 enhanced the effect of AEA; AM251 blocked the effect of WIN-2	the Ang II-VP induced hypertension might be diminished by **eCBs acting at CB_1_Rs**	[46]
**rats** **SHR** **WKY**	conscious		BP was higher in SHR than in WKY	RIM 3 mg/kg i.v. URB597 1.7 mg/kg i.v.	RIM ↑BP and URB597 ↓BP in SHR but not in normotensive WKY	in SHR in which RAS is overactivated **eCBs acting at CB_1_Rs** reduce BP	[19]
**rats**	conscious (mRen2)27 hypertensive rats or normotensive SD	**(mRen2)27**: higher RAS activity		RIM (10 mg/kg; p.o. acutely or daily for 28 days)	**acutely:** ↓BP and ↓HR in hypertensive but not in SD **chronically:** ↓BP and ↓HR; ↔plasma Ang II, ↔Ang 1-7; ↔ACE; improvement of sympathetic and parasympathetic BRS	**upregulated ECS** contributes to hypertension and impaired autonomic function in this Ang II-dependent model; systemic **CB_1_R** blockade may be an effective therapy for **Ang II-dependent hypertension** and the associated metabolic syndrome	[47]
**rats**	anaesthetized (mRen2)27 hypertensive, ASrAOGEN and SD rats	**(mRen2)27:** higher RAS activity; AsrAO-GEN: low glial angiotensinogen	**levels in NTS:** *2-AG*: (mRen2)27 > SD > ASrAOGEN; *AEA*: (mRen2)27 ≈ SD ≈ ASrAOGEN **dorsal medulla**: *CB_1_*: ASrAOGEN < (mRen2)27 ≈ SD; *CB_2_*: no differences	RIM (0.36 and 36 pmol/rat; NTS)	↑BRS in (mRen2)27; ↓BRS in ASrAOGEN; ↔BRS in SD	upregulated **brain ECS in Ang II**-dependent hypertension may contribute to the impaired baroreceptor sensitivity in this model of hypertension	[48]
**rats**	obese fa/fa Zucker rats and control lean fa/+ Zucker rats; isoflurane-anaesthetized	acute Ang II (30 and 100 ng/kg, i.v.)	stronger pressor response in obese than in lean rats	RIM (3 or 10 mg/kg, p.o.) for 12 months	normalized the acute pressor response to Ang II in obese rats to the level of lean rats	authors suggest that **chronic CB_1_R blockade** by RIM might **reduce** vascular **AT_1_R** expression; an indirect mechanism related to the decrease in the cholesterol level should also be taken under consideration	[49]
**rats** **SHR** **WKY**	conscious		SHR in comparison to WKY: higher BP, **carotid, mesenteric artery**: ↑AT_1_Rs, ↑ACE **kidney**: ↔AT_1_Rs, ↔AT_2_Rs, ↑ACE	PEA (30 mg/kg; s.c. for 5 weeks)	BP in SHR↓ **SHR****arteries**: ↓AT_1_Rs, ↓ACE level **SHR kidney****:** ↓AT_1_Rs, ↑AT_2_Rs, and ↓ACE level associated with ↓oxidative and nitrosative stress	PEA lowers BP and protects against hypertensive renal injury partially via reduction in vascular **AT_1_Rs** and Ang II-mediated effects and via **modulation of the RAS, leading the AT_1_/AT_2_ balance** towards an anti-hypertensive status	[50,51]
**rats** **WKY**	cultured lymphocytes from WKY	Ang II (0.01–1)	concentration-dependent ↓AEA transporter activity and ↑ROS level	losartan (10 and 100)	↓Ang II effects on AEA transporter activity and ROS level	Ang II plays a critical role in mediating the decrease in **AEA transporter activity** in SHR; probably via AT_1_Rs	[52]
**rats** **SHR** **WKY**	conscious		SHR: ↑plasma Ang II and ↑AEA level; ↓AEA transporter activity in comparison to WKY	losartan (15 or 30 mg/kg; p.o. for 2 weeks)	restoration of reduced AEA transporter activity; ↓plasma AEA level		
**Nervous System**							
**rats** **Wistar**	urethane- anesthetized	Ang II (0.14 nmol/rat; PVN)	↑BP	AM251 (0.48 nmol/rat; PVN)	AM251 reduced the Ang II-mediated BP increase and slightly increased BP by itself	Ang II-induced hypertension involves CB_1_Rs in the PVN	[53]
**rats** **Wistar**	urethane- anaesthetised (microinjection into the PVN, doses in nmol/rat)	CP55940 (0.1) CP55940 (0.1) + AM251 (3 μmol/kg; i.v.)	↓BP, ↓HR ↑BP, ↑HR	losartan (10 μmol/kg; i.v.) losartan (10 μmol/kg; i.v.)	no effect reversed ↑BP, ↑HR to ↓BP, ↓HR	presynaptic inhibitory CB_1_Rs on GABAergic neurons in the PVN activated by eCBs released in response to Ang II modify the glutamatergic neurotransmission enhanced by presynaptic AT_1_R activation	[54]
**rats** **SHR** **WKY**	conscious (all compounds microinjected into the PVN, nmol/rat)	Ang II (0.03) or CP55940 (0.1) + AM251 (3 μmol/kg; i.v.)	↑BP stronger in SHR than in WKY	losartan (20) PD123319 (10) AM251 (30)	↓pressor effect of Ang II and CP55940 ↓pressor effect of Ang II and CP55940 ↓pressor effect of Ang II	mutual interaction in the PVN between CB_1_Rs and receptors for Ang II responsible for stimulation of the pressor response (probably via stimulation of CB_1_R by eCBs released in response to Ang II)	[55]
**mice**	magnocellular neurosecretory cells from the supraoptic nucleus	Ang II (0.1)	↑frequency of mEPSCs	AM251 (2)	↑effect of Ang II	eCBs released in response to Ang II modulate the excitatory synaptic inputs via negative feedback	[56]
**rats Wistar** **mice** **CB_1_^+/+^** **CB_1_^−/−^**	conscious	Ang II (191 pmol/rat; i.c.v.) Ang II (191 pmol/mouse i.c.v.)	↓ethanol-induced gastric lesions (reduced by candesartan 5.2 and 31.7 nmol/rat; i.c.v.) gastroprotection in CB_1_^+/+^ as opposed to CB_1_^−/−^	AM251 (1.8 nmol/rat; i.c.v.) THL (0.2 nmol/rat; i.c.v.)	inhibition of the gastroprotective effect of Ang II	**Ang II stimulates eCB release via activation of central AT_1_R receptors**, and activation of CB_1_Rs induces gastroprotection in a vagus-mediated mechanism (inhibition by vagotomy and atropine)	[57]
**mice** **CB_1_^+/+^** **CB_1_^−/^**	response of the chorda tympani (CT) nerve in anesthetized mice	CB_1_^+/+^: Ang II (100–5000 ng/kg; i.p.)	gustatory nerve responses ↓ to NaCl and ↑ to sweeteners, blocked by candesartan	CB_1_^–/–^: Ang II (100–5000 ng/kg; i.p.)	gustatory nerve responses ↓ to NaCl and ↔ to sweeteners	enhancing effect of Ang II on sweet taste responses mediated by AT_1_ and CB_1_Rs; authors suggest that **Ang II, via AT_1_Rs, stimulates the release of 2-AG** that may act as an autocrine enhancer for **CB_1_Rs** on sweet taste cells	[58]
**rats** **SHR** **WKY**	**astrocytes** basal CB_1_R densities: brainstem: SHR<WKY cerebellum: SHR>WKY	Ang II (0.1)	SHR: ↓CB_1_R and ↑CB_1_R densities and phosphorylation in brainstem and cerebellar astrocytes, respectively; opposite effects in WKY	losartan (10) PD123319 (10) ACEA (0.01)	- effects of Ang II were inhibited by losartan (brainstem) and by losartan and PD123319 (cerebellum) - ACEA reduced the AT_1_R-mediated MAPK activation in brainstem and cerebellar astrocytes	Ang II, mostly via the AT_1_R, is capable of **altering CB_1_R expression** and phosphorylation in astrocytes isolated from the brainstem and cerebellum under hyper- and normotensive conditions; possible role in neuroinflammatory and attention-deficit hyperactivity disorders, respectively	[59,60]
**rats** **SHR** **WKY**	astrocytes isolated from the brainstem and from cerebellum	Ang II (0.1)	↓IL-10 and ↑IL-1β gene expression in astrocytes from both brain regions of SHR and WKY	ACEA (0.01)	co-treatment of Ang II and ACEA resulted in the neutralization of Ang II-mediated effect in WKY but not SHR	Ang II and ACEA have opposing roles in the regulation of inflammatory gene signature in astrocytes isolated from SHR and Wistar rats (possible **functional antagonism**)	[61]
**mice** **CB_2_^−/−^** **CB_2_^+/+^**	hippocampus slices		CB_2_^−/−^: ↓ACE level, and ↑aβP in comparison to WT		CB_2_R deletion:↑aβ neurotoxicity associated with ↓level of ACE (that degrades aβ)	**activation of CB_2_Rs increases ACE level** that degrades aβ; possible significance in Alzheimer’s disease	[62]
	N2a cells overexpressing aβP	JWH133	↑ACE level, ↓aβP	AM630	all JWH133 effects were attenuated		
**Kidney**							
**humans**	podocytes	Ang II (0.1)	↑AEA, ↑2-AG ↑AT_1_Rs and CB_1_Rs	JD5037 (100) or losartan (10)	↓ all changes induced by Ang II	**peripheral CB_1_R** blockade might possess therapeutic potential in disease(s) connected with enhanced RAS	[63]
**rats**	Zucker diabetic fatty rats with nephropathy; control lean rats	diabetic compared to lean rats	↑plasma Ang II and aldosterone levels; ↓AT_1_Rs in renal cortex	JD5037 (3 mg/kg p.o. for 3 months) losartan (20 mg/kg p.o. for 28 days)	↓plasma Ang II and aldosterone levels; ↓AT_1_Rs in renal cortex ↔plasma Ang II and ↓aldosterone levels; ↓CB_1_Rs in renal cortex		
**mice**	streptozotocin- induced diabetic nephropathy		↑glomerular CB_1_ and ↑AT_1_Rs; ↔CB_2_Rs	AM6545 (10 mg/kg; i.p.) alone or together with perindopril (2 mg/kg; p.o.) for 14 weeks	**Single treatments** ↔glomerular CB_1_-, CB_2_Rs, and ↓AT_1_Rs; ↓progression of albuminuria, down-regulation of nephrin and podocin, ↓inflammation, and ↓expression of markers of fibrosis **Combined****treatment** ↔glomerular CB_1_-, CB_2_Rs and ↓AT_1_Rs; also reversal of albuminuria	The superior effect of **dual therapy (peripheral CB_1_R antagonist + ACE inhibitor)** on albuminuria, nephrin loss, and inflammation suggest that CB_1_R blockade may be a valuable option as an additional therapy, although single and combined treatment only reduce glomerular AT_1_Rs without affecting CB_1_Rs and CB_2_Rs.	[64]

↓—decrease; ↑—increase; ↔—no change. 2-AG, 2-arachidonoyl glycerol; A549, alveolar epithelial cell line; AβP, amyloid-β protein; ACE, angiotensin-converting enzyme; ACE2, angiotensin-converting enzyme 2; ACEA, arachidonyl-2’-chloroethylamide; AEA, anandamide; Ang II, angiotensin II; Ang 1-7, angiotensin 1-7; ApoE, apolipoprotein E; ASrAOGEN, transgenic rats characterized by a transgene producing antisense RNA against angiotensinogen in the brain; AT_1_R, Ang II receptor type 1; AT_2_R, Ang II receptor type 2; BRS, baroreceptor sensitivity; CB_1_R, cannabinoid receptor type 1; CB_2_R, cannabinoid receptor type 2; CBD, cannabidiol; CBG, cannabigerol; CBN, cannabinol; CF, coronary flow; CHO, Chinese hamster ovary cells; DAGL, diacylglycerol lipase; eCBs, endocannabinoids; ECS, endocannabinoid system; EHP-101 (VCE-004.8), oral lipidic formulation of the novel non-psychotropic cannabidiol aminoquinone; ERK, extracellular signal-regulated kinases; FAAH, fatty acid amide hydrolase; hACE2, human ACE2; hiPSC-CMs, human iPSC-derived cardiomyocytes; HSC, hepatic stellate cells; IFN-γ, interferon γ; i.c.v., intracerebroventricular; IL-1β, interleukin-1β; IL-10, interleukin-10; i.p., intraperitoneal; i.v., intravenous; HR, heart rate; LDH, lactate dehydrogenase; LV, left ventricle; MAGL, monoacylglycerol lipase; MAPK, mitogen-activated protein kinase; mEPSCs, miniature excitatory postsynaptic currents; (mRen2)27, Ang II-dependent hypertension model; NA, noradrenaline; NTS, solitary tract nucleus; PEA, *N*-palmitoylethanolamide; pERK, phospho-ERK; p.o., per os; PVN, paraventricular nucleus of hypothalamus; RIM, rimonabant; RAS, renin angiotensin system; ROS, reactive oxygen species; s.c., subcutaneous; SD, Sprague-Dawley rats; SHR, spontaneously hypertensive rat; TgA, transgenic rat, model of preeclampsia; THC, Δ⁹-tetrahydrocannabinol; THCV, tetrahydrocannabivarin; THL, tetrahydrolipstatin; TMPRSS2, transmembrane serine protease 2; TNF-α, tumor necrosis factor α; URB597, an inhibitor of FAAH (fatty acid amide hydrolase); WIN-2, WIN55212-2; WKY, Wistar Kyoto rats; WT, wild type; VSMCs, vascular smooth muscle cells; VP, vasopressin.

#### 3.2.1. Angiotensin II Causes Transactivation of Cannabinoid-Mediated Effects

The AT_1_Rs for Ang II interact with various G proteins, including G_q/11_ [53,65,66]. Activation of G_q/11_ protein-coupled receptors and the subsequent stimulation of phospholipase C leads to Ca^2+^ signal generation and indirectly causes rapid biosynthesis of eCBs (mainly 2-AG) by DAGL activation (Figure 4). As mentioned above, 2-AG exerts its effects mainly via CB_1_Rs and CB_2_Rs [8]. This reasoning suggests that Ang II might cause transactivation of cannabinoid-mediated effects.

The following facts determined by Turu et al. [33] on Chinese hamster ovary (CHO) cells co-expressing CB_1_Rs and AT_1_Rs clearly confirm the above working hypothesis. Firstly, Ang II increased levels of 2-AG (measured by mass spectrometry) but not of AEA. Secondly, Ang II elicited G_o_ protein activation (determined by detecting the dissociation of G protein subunits with bioluminescence resonance energy transfer; BRET) that was blocked both by the CB_1_R antagonist AM251 and the DAGL inhibitor tetrahydrolipstatin (THL); AM251 and THL by themselves inhibited basal CB_1_R activity. The pioneering observations by Turu et al. [33] have been confirmed and extended by the same group on HEK293 and COS7 cells from kidney and fibroblasts, respectively [34]. Thus, Ang II caused CB_1_R-dependent activation not only in an autocrine but also in a paracrine manner [34], i.e., when AT_1_Rs and CB_1_Rs are localized on the same or separate cells, respectively.

The latter interaction obtained on cell culture has been verified in blood vessels and other body systems. First of all, Szekeres et al. [36] showed in vitro that Ang II enhanced the calcium signal and the 2-AG level in rat aortic vascular smooth muscle cells (VSMCs). Both responses to Ang II were reduced by the DAGL inhibitor THL and enhanced by the MAGL inhibitor JZL184 leading to inhibition of 2-AG synthesis and degradation and finally to a decrease and increase in 2-AG levels, respectively. The results obtained on cell culture were further confirmed on isolated vessels examined in the organ bath, including the wire myograph for studies on small resistance arteries. As shown in Table 2, Ang II caused a concentration-dependent vasoconstriction of rat and/or mice aortic rings [36], skeletal muscle arterioles and saphenous arteries [37], and intramural coronary resistance arterioles [38]. All vasoconstrictory responses to Ang II were enhanced by antagonists of CB_1_Rs (the inverse agonists rimonabant and AM251 or the neutral antagonist O2050) and by the DAGL inhibitor THL via reduction of the CB_1_R-mediated vasodilatation or 2-AG levels, respectively. On the other hand, the MAGL inhibitor JZL184, increasing 2-AG levels, reduced the above effects of Ang II. Moreover, the CB_1_R antagonists had an amplificatory influence only in wild-type but not in CB_1_R knockout mice [36,37]. The presence of CB_1_Rs in the particular vessels or aortic VSMC has been confirmed with immunohistochemistry or using the Western blot technique [36,37,38] or functionally by studying the vasodilatory effect of the CB_1_R agonist WIN55212-2 with or without CB_1_R blockade [36,37,38]. Moreover, WIN55212-2-induced vasodilatation was not observed in CB_1_^−/−^ mice [36,37].

An interaction between Ang II and CB_1_Rs resembling that in systemic arteries (see above) was also observed in rat pulmonary arteries, in which CB_1_Rs are present in smooth muscle, as determined using immunohistochemical staining [39]. Thus, the Ang II-induced concentration-dependent contraction of pulmonary arteries was amplified by AM251 and the DAGL inhibitor RHC8026 and diminished by inhibition of 2-AG but not AEA degradation by JZL184 and URB597, respectively. Unfortunately, the Ang II-evoked response in human pulmonary arteries was very weak and did not allow for examining its potential interaction with CB_1_-Rs [39].

In uterine arteries isolated from control and TgA rats (model that mimics many features of preeclampsia), the vasoconstrictor responses to Ang II (which were stronger in TgA than in control animals) were decreased by inhibitors of FAAH (URB597) and/or MAGL (JZL184) but not by the CB_1_R antagonist rimonabant [40]. Authors suggest that eCBs released in response to Ang II reduced the Ang II-induced contraction in a CB_1_R- independent manner. Indeed, the non-selective COX inhibitor indomethacin reduced the maximal contraction and sensitivity to Ang II in arteries from TgA rats, suggesting that eCBs (both AEA and 2-AG) released by Ang II were quickly converted to vasoactive eicosanoids via COX, yielding vasodilator prostaglandins and/or prostacyclin. 

In summary, the above in vitro experiments on isolated arteries clearly show the existence of protective negative feedback in response to the vasoconstrictory effect of Ang II. Thus, Ang II acting via AT_1_Rs not only contracts particular arteries but also stimulates a DAGL-dependent release of eCBs (mainly 2-AG) from vascular endothelium or smooth muscles that cause vasodilatation in a CB_1_R-dependent (Figure 4) or independent manner, thereby diminishing the initial response to Ang II.

An interaction between AT_1_Rs and eCBs is also suggested to be present in the heart ([42]; Table 1, Figure 4). In rat isolated Langendorff-perfused hearts Ang II reduced coronary flow (CF) and caused a moderate negative inotropic effect, although it is known to exert a positive inotropic effect in other heart preparations (e.g., on rat cardiomyocytes in the study by Rajagopal et al. [67]). In Langendorff-perfused hearts, the activation of CB_1_Rs (the presence of which was confirmed by immunohistochemistry) by WIN55212-2 and 2-AG increased CF and decreased cardiac contractility in a manner sensitive to the CB_1_R antagonist O2050. However, in contrast to previous experiments on rat coronary arteries [38], O2050 and the DAGL inhibitor orlistat did not amplify but reduced both cardiac effects of Ang II. Miklós et al. [42] argue that the final response to Ang II in the heart is determined by an interplay between coronary arteries and cardiomyocyte contractility, and the degree of CF results primarily from the cardiac oxygen demand and not only from the direct vascular effects of vasoactive agents. Authors suggest that CB_1_R blockade by O2050 prevented the CB_1_R-induced negative inotropic effect elicited by 2-AG released under Ang II stimulation. Ang II then increases cardiac contractility (otherwise masked by flow deprivation) and, subsequently, the overall oxygen demand and the CF. 

The question arises whether the protective negative feedback of Ang II—eCBs—CB_1_Rs observed in vitro is relevant also under in vivo conditions. The answer is “yes”, as shown in Table 1. First of all, Szekeres et al. [36] showed that, as in experiments on isolated arteries, the increase in blood pressure (BP) elicited by intravenous (i.v.) infusion of Ang II was enhanced by the CB_1_R antagonist O2050 in anesthetized CB_1_^+/+^ R wild type as opposed to CB_1_^−/−^ R knockout mice. A similar mechanism may occur in experiments performed on conscious rats with hypertension induced by infusion of Ang II [19] or Ang II plus vasopressin [46]. Thus, the CB_1_R antagonist AM251 enhanced and the FAAH inhibitor URB597 reduced BP [19], and URB597 enhanced the fall in BP elicited by AEA [46]. Similarly, in spontaneously hypertensive (SHR) rats (in which the RAS is overactivated [68]) but not in their normotensive Wistar Kyoto (WKY) controls, the CB_1_R antagonist rimonabant and the FAAH inhibitor URB597 enhanced and reduced BP, respectively [19]. The blockade of the hypotensive effect of AEA and WIN55212-2 by AM251 confirms the involvement of CB_1_Rs in the hypotensive action of both compounds [46].

Interestingly, an opposite effect of CB_1_R blockade was obtained in transgenic (mRen2)27 rats, i.e., a monogenetic model of Ang II-dependent hypertension in which the mouse Ren2 renin gene was transfected into the genome of the Sprague-Dawley (SD) rat. (mRen2)27 rats have the phenotype of chronic hypertension with markedly impaired baroreflex control over heart rate (HR) and increased body weight. Unlike in SHR or hypertension models induced by Ang II [19] or Ang II with vasopressin infusion [46], acute or chronic administration of rimonabant decreased BP and body weight in (mRen2)27 rats but not in their normotensive SD controls [47]. One can exclude that the reduction of body weight is responsible for the fall in BP since it resulted not only from chronic but also from acute administration of the CB_1_R antagonist. Moreover, chronic rimonabant administration improved sympathetic and parasympathetic baroreflex sensitivity for HR control but failed to change plasma Ang II and Ang 1-7 levels or ACE activity. The authors concluded that an upregulated endocannabinoid system contributes to hypertension and impaired autonomic function in this Ang II-dependent model and that systemic CB_1_R blockade may be an effective therapy in this type of hypertension [47]. In a subsequent study, Schaich et al. [48] confirmed their conclusion showing enhanced levels of 2-AG (but not AEA) in the solitary tract nucleus (NTS) of (mRen2)27 rats in comparison to SD or ASrAOGEN rats (the latter have a low level of glial angiotensinogen). Microinjection of rimonabant into the NTS dose-dependently enhanced baroreceptor sensitivity in (mRen2)27 but decreased it in ASrAOGEN animals. Thus, upregulation of the brain endocannabinoid system may be responsible for the impairment of baroreceptor sensitivity in this Ang II-dependent model of hypertension.

In this context, one should keep in mind that Ang II administered centrally and peripherally increases BP [69], whereas cannabinoids can elicit different cardiovascular responses (including hypotension and hypertension) [70,71]. Interactions between the (endo)cannabinoid and RAS systems may not only occur in vessels but also in the CNS. The paraventricular nucleus of the hypothalamus (PVN) is one of the crucial brain regions involved in the regulation of cardiovascular functions, in which the balance between sympathoexcitatory glutamatergic and sympathoinhibitory GABAergic transmission is responsible for the final integration of the sympathetic outflow [72]. Gyombolai et al. [53] were the first to show that microinjection of the CB_1_R antagonist AM251 into the PVN of anesthetized rats reduced the increase in BP elicited by application of Ang II to the same nucleus, although the underlying mechanism has not been clarified. We found that the cannabinoid receptor agonist CP55940 given into the PVN of rats anesthetized with urethane decreased BP but in the presence of AM251 given i.v. produced a clear pressor response, sensitive to AM251 given into the PVN [54]. The AT_1_R antagonist losartan given i.v. inhibited the depressor response to CP55940 only but reversed its pressor response to a fall in BP. In our most recent study, performed on conscious SHR and their normotensive (WKY) controls in which all compounds were microinjected into the PVN, the pressor responses to Ang II and to CP55940 (both of which were stronger in SHR than in WKY) were inhibited by antagonists of AT_1_ and AT_2_ receptors, losartan and PD123319, respectively. On the other hand, AM251 diminished the increase in BP induced by Ang II [55]. The pressor effects of Ang II and CP55940 given into the PVN result mainly from the activation of facilitatory AT_1_Rs on glutamatergic [73,74] and inhibitory CB_1_Rs on GABAergic [54,55] neurons, respectively (Figure 5). In addition, Ang II may lead to the release of eCBs that, via CB_1_Rs, reduce the GABAergic tone and eventually lead to increased activity of the glutamatergic neurons associated with a pressor effect. On the other hand, AM251 blocking presynaptic CB_1_Rs on GABAergic neurons may increase the GABAergic inhibitory influence on the sympathoexcitatory glutamatergic tone stimulated by Ang II and, in this way, reduces its pressor response (Figure 5). The potential mechanism responsible for the interaction of AT_2_Rs (which, unlike the AT_1_Rs, are not coupled to G_q/11_ proteins; [75]) with CB_1_Rs remains to be established. AT_2_Rs are known mainly to counteract the pressor effect of AT_1_R stimulation by enhancing GABAergic neurotransmission (AT_2_Rs are located predominantly on GABAergic neurons; [76]).

Interestingly, the mechanism related to the Ang II-induced transactivation of the cannabinoid-mediated effects is not restricted to the cardiovascular system. Thus, AM251 enhanced the Ang II-induced miniature excitatory postsynaptic currents (mEPSCs) in supraoptic magnocellular neurons from the supraoptic nucleus (SON), which is crucial for body fluid homeostasis (Table 2, Figure 4). The results suggest that eCBs released by Ang II modulate the excitatory synaptic inputs via presynaptic inhibitory CB_1_Rs [56]. The authors based their conclusion on the results of their previous publication, in which they proved that CP55940 reduced the frequency of spontaneous mEPSCs in magnocellular neurosecretory neurones of the SON in a manner sensitive to AM251 [77]. On the other hand, there may be an alternative explanation for the interaction. Thus, CB_1_Rs are pre-coupled receptors, and an inverse agonist like AM251 is expected to lead to an increase in mEPSCs even in the absence of eCBs. Thus, Ang II and AM251 may increase mEPSCs via two independent mechanisms, i.e., via agonism at facilitatory AT_1_Rs and inverse agonism at inhibitory CB_1_Rs, respectively.

Gyires et al. [57] showed that the gastroprotective effect of Ang II given intracerebroventricularly (i.c.v.) against ethanol-induced gastric lesions in conscious rats was reduced not only by the AT_1_R antagonist candesartan but also by AM251 and the 2-AG synthesis inhibitor THL (all compounds were injected i.c.v.). Moreover, the beneficial influence of Ang II was absent in CB_1_^−/−^ knockout mice. The authors suggest that Ang II caused gastroprotection via stimulation of central AT_1_Rs that led to the release of eCBs in a DAGL-dependent manner and the subsequent activation of CB_1_Rs (Figure 4). CB_1_Rs, like AT_1_Rs, are known to exert beneficial effects on gastrointestinal function [57].

Shigemura et al. [58] proved that intraperitoneal (i.p.) administration of Ang II decreased the response of the gustatory nerve to NaCl both in the wild type and in CB_1_^−/−^ knockout mice. By contrast, the genetic deletion of CB_1_Rs prevented the enhancement of the response to sweeteners induced by Ang II. Authors suggested that activation of AT_1_Rs by Ang II stimulates 2-AG release, which may act as an autocrine enhancer for CB_1_Rs on sweet taste cells (Table 2, Figure 4). The CB_1_R-dependent enhancement of gustatory nerve responses to sweeteners induced by AEA or 2-AG had previously been demonstrated in mice by Yoshida et al. [78].

The authors of all the above publications argue that Ang II causes the release of eCBs (mainly 2-AG). Unfortunately, as shown in Table 2, this interesting hypothesis has been confirmed directly by determining eCB levels in cell culture and vessels only [33,36]. In the other studies it is based mainly on indirect proof using inhibitors of eCB synthesis and/or degradation and CB_1_R antagonists.

#### 3.2.2. Other Forms of Interaction between Angiotensin II and Cannabinoid-Mediated Effects

Numerous authors [43,47,48,53,54,63,64] argue that the interaction between CB_1_ and AT_1_ receptors they found is based on AT_1_R-CB_1_R heteromerization. However, this has so far been proven only in the excellent study by Rozenfeld et al. [35], in which coimmunoprecipitation, resonance energy transfer assays, and receptor- and heteromer-selective antibodies were used. The authors demonstrated that G_αq_-coupled AT_1_Rs and G_αi_-coupled-CB_1_Rs form receptor heteromers in cells in which both receptors occur naturally (hepatic stellate cells [HSCs] isolated from ethanol-administered rats), and naturally (CB_1_) and recombinantly (AT_1_) (cultured Neuro 2A cells) (Table 2). The following results on cultured Neuro 2A cells co-expressing CB_1_ and AT_1_ receptors clearly confirm the existence of AT_1_R-CB_1_R heteromerization: (1) Ang II-induced an increase in ERK1/2 phosphorylation via G_αi_ (for CB_1_Rs) instead of G_αq_ (for AT_1_Rs), which was reduced by decreasing CB_1_R levels by siRNA, and by rimonabant or THL; (2) the CB_1_R agonist HU210 stimulated ERK1/2 phosphorylation only in the presence of a very low non-signaling concentration of Ang II; (3) an increase in intracellular Ca^2+^ levels is induced by activation of AT_1_ but not CB_1_ receptors, yet requires the presence of CB_1_Rs since it was attenuated by siRNA targeting CB_1_Rs; (4) the expression of AT_1_Rs shifted the localization of CB_1_Rs from an intracellular compartment to the plasma membrane. The physiological relevance of AT_1_R-CB_1_R heteromerization has been demonstrated in HSCs isolated from ethanol-treated rats and their controls. Firstly, the levels of CB_1_Rs, AT_1_Rs, and AT_1_R-CB_1_ heteromers were upregulated in ethanol-treated HSCs compared to controls. Secondly, the CB_1_R antagonist rimonabant prevented Ang II-mediated mitogenic signaling and profibrogenic gene expression. The authors concluded that enhanced CB_1_R expression in activated HSCs might affect AT_1_R properties and contribute to the profibrogenic effect of Ang II.

Attenuation of the AT_1_R-mediated effect by CB_1_R blockade is likely for the postganglionic sympathetic neurons, and AT_1_R-CB_1_R heteromerization might explain this phenomenon. In general, sympathetic neurons are equipped with presynaptic inhibitory G_i/o_ protein- and facilitatory G_q_ and G_s_ protein-coupled receptors. Although G_i/o_ and G_s_ protein-coupled receptors in the mouse atrium act independently [79], this does not hold for the interplay between the G_i/o_ and G_q_ protein-coupled receptors [79,80]. Thus, if the presynaptic inhibitory α_2_-adrenoceptor (an example of a G_i/o_ protein-coupled receptor) is blocked by an α-adrenoceptor antagonist, the AT_1_R-mediated facilitation of noradrenaline release from the sympathetic neurons is impaired [79]. Facilitation can be restored if an agonist of another G_i/o_ protein-coupled receptor is added, e.g., a δ-opioid receptor agonist or neuropeptide Y (acting via presynaptic Y_2_ receptors). The same occurs if the presynaptic bradykinin (B_2_) receptor is considered instead of the AT_1_R [79]. In harmony with these findings, AT_1_R (and B_2_ receptor)-mediated facilitation of noradrenaline release is impaired in α_2_-adrenoceptor knockout mice [80]. Unfortunately, the interaction between the presynaptic AT_1_R and CB_1_R has not been examined so far.

In a few studies, antagonists of CB_1_Rs caused down-regulation of AT_1_Rs and subsequently also the pathological effects elicited by Ang II. Such changes have been observed in vitro and in vivo following acute and chronic administration of CB_1_R antagonists. Thus, in aortic VSMCs, rimonabant and AM251 (but not the CB_2_R antagonist SR144528) reduced the oxidative stress induced by Ang II. Moreover, blockade by CB_1_Rs or their stimulation by CP55940 led to down- and up-regulation of AT_1_Rs, respectively. The level of CB_1_Rs was not affected by rimonabant in VSMCs ([41]; Table 1). Chronic administration of rimonabant for 7 weeks to apolipoprotein E-deficient (ApoE^−/−^) mice on a cholesterol-rich diet did not prevent atherosclerotic plaque development, collagen content, and macrophage infiltration but, like in the above in vitro experiments, reduced oxidative stress and improved aortic endothelium-dependent vasodilation associated with the down-regulation of AT_1_Rs [41]. 

Similarly, application of the CB_1_R antagonist rimonabant or AM281 for 11 weeks or genetic deletion of CB_1_Rs in mice reduced AT_1_ and CB_1_ receptor levels in the left cardiac ventricle and improved cardiac dysfunction in streptozotocin-induced diabetes ([43] Table 2). Moreover, treating obese Zucker rats with rimonabant for 12 months reduced the pressor response to acute i.v. injection of Ang II in comparison to vehicle-treated control animals. The CB_1_R blockade also improved renal function and metabolic profile and increased the survival of obese Zucker rats ([49], Table 2). The authors suggested that rimonabant might reduce the vascular AT_1_R expression. However, an indirect mechanism related to the decrease in cholesterol level should also be considered.

The paper by Jourdan et al. ([63]; Table 2) refers to the interaction between AT_1_ and CB_1_ receptors in kidneys. In an in vitro study on human podocytes, the peripheral CB_1_R antagonist JD5037 and the AT_1_R antagonist losartan diminished the Ang II-induced enhancement of AEA and 2-AG levels and the increased expression of both CB_1_ and AT_1_ receptors. These results are in line with in vivo observations on Zucker diabetic fatty rats with nephropathy, which had higher plasma levels of Ang II and aldosterone as well as up-regulated AT_1_ and CB_1_ receptors in the renal cortex in comparison with control lean rats [63]. Chronic administration of JD5037 for 3 months or losartan for 4 weeks prevented/attenuated the development of nephropathy and decreased the expression of AT_1_ and CB_1_ receptors in the renal cortex. Authors suggest that activation of renal CB_1_Rs aggravates the deleterious effects of RAS activity. Indeed, in rats with streptozotocin-induced diabetes, dual 14-week therapy with the peripheral CB_1_R antagonist AM6545 and the ACE inhibitor perindopril was more effective than monotherapies. Thus, single antagonists produced various beneficial effects but only combined treatment very markedly reduced albuminuria ([64]; for details, see Table 2). Importantly, diabetic nephropathy was associated with higher glomerular levels of CB_1_ and AT_1_ receptors. Both mono- and combined therapy failed to modify glomerular CB_1_R expression but normalized AT_1_R levels. Altogether, the studies by Jourdan et al. [63] and Barutta et al. [64] suggest that both RAS inhibition and CB_1_ receptor blockade have a beneficial influence on the locally increased responsiveness to Ang II.

The studies by Mattace Raso et al. [50,51] and Franco-Vadillo et al. [44] document a beneficial and CB_1_R-independent influence of two cannabinoids on the harmful effects of Ang II (for details, see Table 1). A decrease in AT_1_R densities was found in the heart [44], carotid and mesenteric arteries [51], and kidney [50] in response to chronic administration of CBD for 10 days to rats [44] or PEA for 5 weeks to hypertensive SHR and their normotensive WKY counterparts [50,51]. CBD or PEA treatment also reduced detrimental changes in cardiac [44], vascular and renal [50,51] tissues, and BP in SHR [50,51]. CBD and PEA show an overlap in their biochemical effects [81]. CBD is the major nonpsychoactive cannabinoid constituent of marijuana and has a low affinity for CB_1_ and CB_2_ receptors (Figure 2; [82]). PEA belongs to so-called endocannabinoid-like compounds with similar chemical structures to eCBs but is devoid of affinity for any CB receptor [8]. Apart from regulating the endocannabinoid system, CBD has a complex pharmacodynamic profile, including PPARγ [82]. The primary mechanism of action of PEA is its direct activation of PPARα receptors [81]. PPARγ agonists are known to decrease BP in humans, possibly through the suppression of the RAS, including the inhibition of AT_1_R expression [83]. Similarly, activation of PPARα decreases the Ang II-mediated rise in BP [84]. Thus, PPARs may represent the intermediate link in the interaction between CBD, PEA, and RAS, although this hypothesis needs to be proven by further studies. Of course, other explanations of the influence of CBD and PEA on AT_1_R densities are possible, especially when considering the pleiotropic action of CBD. 

Beneficial effects against Ang II-induced fibrosis and inflammation in mouse heart, aorta, lung, kidney, and skin were obtained after chronic administration of EHP-101, an oral lipidic formulation of the novel non-psychotropic cannabidiol aminoquinone VCE-004.8, a dual agonist of CB_2_R and PPARγ receptors ([45]; Table 2). Both properties may contribute to a functional antagonism against the detrimental consequence of AT_1_R stimulation.

An opposite cross-talk, i.e., the ability of Ang II to reduce CB receptor densities, has been identified in brainstem and cerebellar astrocytes that play critical roles in hypertension and attention-deficit hyperactivity disorder, respectively, and in which activation of astroglial AT_1_Rs and CB_1_Rs has been determined to increase and to neutralize the inflammatory state, respectively ([59,85]; for details, see Table 2). Authors compared the effects of Ang II on CB_1_Rs in astrocytes isolated from hypertensive SHR and normotensive WKY rats [59]; they found that basal expression of CB_1_Rs in brainstem astrocytes was lower and that in cerebellar astrocytes was higher in SHR than in WKY rats. Ang II infusion further decreased (brainstem) and increased (cerebellum) CB_1_R expression via AT_1_Rs (brainstem) and both AT_1_Rs and AT_2_Rs (cerebellum). In further studies, Haspula and Clark [60] showed that stimulation of CB_1_Rs by their potent agonist ACEA led to inhibition of AT_1_R-mediated MAPK activation. On the other hand, Ang II caused phosphorylation and, probably, inactivation of CB_1_Rs. Moreover, ACEA caused the neutralization of Ang II-mediated changes in IL-10 and IL-1β gene expression in astrocytes from both brain regions of SHR and WKY [61].

Interesting results were obtained by Shi et al. [52], who found that Ang II decreased the activity of the AEA transporter in cultured lymphocytes from WKY in a manner sensitive to the AT_1_R antagonist losartan; this in vitro observation was confirmed under in vivo conditions in SHR and WKY (Table 2). Plasma Ang II and AEA levels were higher in SHR than in WKY. The higher AEA level probably resulted from the lower activity of the AEA transporter in SHR. Chronic administration of losartan not only dose-dependently reduced BP but also increased AEA transporter activity and diminished plasma AEA levels. The authors concluded that Ang II plays a critical role in mediating the decrease in AEA transporter activity in SHRs, probably via AT_1_Rs [52].

Importantly, there are more and more examples that cannabinoids modulate ACE levels or activity. Wang et al. [62] were probably the first to show that the CB_2_R agonist JWH133 increased ACE levels in Neuro-2a cells in a manner sensitive to the CB_2_R antagonist AM630. ACE levels were also lower in hippocampus slices isolated from knockout CB_2_^−/−^ mice in comparison to the wild-type animals (Table 2). Further examples of the influence of cannabinoids on ACE and, in particular, on ACE2 are given in part 3.4.

#### 3.2.3. Typology and Potential Therapeutic Significance of the Cross-Talk between (Endo) Cannabinoids and the Renin-Angiotensin System

A detailed analysis of the results shown in Table 2, based on 33 papers, reveals that the cross-talk between the (endo) cannabinoid and renin-angiotensin systems mainly occurs in the following two ways (a. and b.).

a. Stimulation of AT_1_Rs by Ang II induces the release of eCBs (mainly 2-AG), which acts at CB_1_Rs, thereby modifying the final effects of Ang II. eCB release by Ang II was either determined directly in various cultured cells [33,34], aortic VSMCs [36], and human podocytes [63], or indirectly in isolated blood vessels [36,37,38,39,40,41], isolated heart [42], or conscious rats [19,46] by studying the effects of inhibitors of eCB synthesis or degradation. CB_1_R antagonists modified the effects induced by AT_1_R activation both under in vitro conditions in cell culture [33,34], human podocytes [63], isolated blood vessels [36,37,38,39,41] isolated heart [42], magnocellular neurosecretory cells from the supraoptic nucleus [56], and under in vivo conditions in conscious and/or anaesthetized rats or mice [19,46,54,55,57,58].

Depending on the model (Table 2 and Figure 4), CB_1_R blockade or genetic deletion of CB_1_Rs may enhance or reduce the effects elicited by AT_1_R activation. The final result of the cross-talk between AT_1_ and CB_1_ receptors depends on whether CB_1_R activation induces an effect in the same or the opposite direction compared to AT_1_R stimulation. In the case of opposite effects mediated via CB_1_ and AT_1_ receptors, CB_1_R antagonists will increase the AT_1_R-mediated effects. This holds for some blood vessels of rodents [36,37,38,39], the rat heart [42], and mEPSCs in magnocellular neurosecretory cells from the supraoptic nucleus of mice [56]. On the other hand, CB_1_R antagonists will reduce the AT_1_R-mediated effects if CB_1_ and AT_1_ receptor activation elicit responses in the same direction (Table 2; Figure 4). This holds for the pressor effect of Ang II after its application into the PVN of the rat [53,54,55], the gastroprotective influence of Ang II given i.c.v. to rodents [57], and the responses to sweeteners in mice [58] (Table 2; Figure 4). 

b. A decrease in AT_1_R expression due to CB_1_R antagonism has been shown less frequently. Thus, acute application of CB_1_R antagonists reduced AT_1_R levels in rodent VSMCs [41] and human podocytes [63], and their chronic administration diminished the AT_1_R expression in the aorta of ApoE^−/−^ mice [41], mouse heart [43], and kidney [63,64] isolated from diabetic mice and/or rats (Table 2; Figure 4). A decrease in AT_1_Rs was also observed in response to chronic treatment of rats with cannabidiol (heart [44]) and PEA (kidney, arteries [44,51]). 

So far, the practical relevance of the cross-talk between ECS and RAS for humans cannot be appreciated. In this respect, CB_1_R antagonists are interesting and are discussed as future drug strategies for the treatment of diabetes [86,87], chronic kidney disease [88], fatty liver disease [89], alcohol use disorder [86], and COVID-19 [90]. The CB_1_R antagonist rimonabant was used as an anti-obesity drug [15,86,91] but withdrawn from the market in 2008 because of severe central side effects [86]. Second-generation CB_1_R antagonists (e.g., AM6545, JD5037), peripherally restricted, may offer advantages over rimonabant. In this context, detailed studies are necessary to clarify whether they increase vascular and cardiac harmful AT_1_R-mediated responses or, by contrast, reduce the detrimental responses to AT_1_R stimulation or expression of AT_1_Rs. One should also keep in mind that FAAH and MAGL inhibitors, which might become future drugs for the treatment of pain, traumatic brain injury, multiple sclerosis, Parkinson’s disease, anxiety-related disorders, and depression [8], may also modify the responses induced by AT_1_R activation (Table 2).

### 3.3. Interaction between Angiotensin 1-7 and (Endo) Cannabinoid-Mediated Effects

So far, only two publications show that not only Ang II but also Ang 1-7 seems to modulate cannabinoid-related effects (Table 3). In the first, Brosnihan et al. [92] showed that the infusion of Ang 1-7 altered the endocannabinoid system in the decidualized uterus of pseudopregnant rats. A Low AEA level is preferred for fertilization, implantation, decidualization, and placentation, whereas a high AEA level causes embryotoxicity. Cannabinoid CB_1_R activation impaired decidualization. The role of 2-AG and CB_2_Rs is still not established in early pregnancy [92,93]. Chronic infusion of Ang 1-7 for 5 days caused an up-regulation of CB_1_R, CB_2_R, and MAGL mRNA in the decidualized uterine horn. However, as decidualization seems to be inhibited by activation of CB_1_Rs, a higher expression level may result in implantation failure, spontaneous miscarriage, or preeclampsia [93].

In the second publication, we identified the cross-talk between MasRs for Ang 1-7 and CB_1_Rs in the PVN of conscious hypertensive SHR and their normotensive controls, WKY ([55]; Table 3). Microinjection of Ang 1-7 or the CB_1_R agonist CP55940 into the PVN produced pressor responses, which were stronger in SHR than in WKY. (Note that CP55940 was microinjected after i.v. administration of the CB_1_R antagonist AM251 to unmask the pressor effect of CB_1_R activation in the PVN; [94]). The increase in BP induced by Ang 1-7 was not only prevented by the MasR antagonist A-779, but surprisingly reversed into a hypotensive response with AM251. On the other hand, the pressor response to CP55940 was reduced not only by AM251 injected into the PVN [94] but also by A-779. Additionally, A-779 decreased BP by itself, but only after the previous i.v. administration of AM251, suggesting a CNS-based interaction between eCBs acting on CB_1_Rs with the endogenous pressor tone elicited by Ang 1-7.

In summary, two in vivo studies on rats suggest that a cross-talk between Ang 1-7 (and its MasRs) and the ECS appears to exist. So far, there are no results/suggestions regarding the mechanism (s) underlying this cross-talk and its potential therapeutic consequence (s).

**Table 3 ijms-23-06350-t003:** Examples of the cross-talk between the (endo)cannabinoid and angiotensin 1-7.

Species	Model	Basal Treatment (Concentration (μM) or Dose)	Effect	Intervention (Concentration (μM) or Dose)	Influence on the Agonist Effect	Final Conclusion of the Authors	References
rats	ovariectomized pseudopregnant	steroid treatment and bolus of oil leading to decidualization	**uterus: ** ↓CB_1_Rs, ↑CB_2_Rs, ↔MAGL, ↓FAAH	Ang 1-7 (24 μg/kg/h; i.u. for 5 days)	**uterus: ** ↑CB_1_Rs, ↑CB_2_Rs, ↑MAGL, ↔FAAH	Ang 1-7 augments the expression of CB_1_Rs, CB_2_Rs, and MAGL in the decidualized uterus and thus may interfere with early events of decidualization	[92]
rats SHR WKY	conscious (compounds microinjected into the PVN, doses in nmol/rat; if not stated otherwise)	Ang 1-7 (0.03) or CP55940 (0.1) + AM251 3 μmol/kg i.v.	both treatments: ↑BP stronger in SHR than in WKY	A-779 (3) AM251 (30)	↓pressor effect of Ang 1-7 and CP55940 ↓pressor effect of Ang 1-7	mutual interaction in the PVN between the CB_1_Rs and the receptors for Ang 1-7 responsible for stimulation of the pressor response	[55]

↓—decrease; ↑—increase; ↔—no change; Ang 1-7, angiotensin 1-7; BP, blood pressure; CB_1_R, cannabinoid receptor type 1; CB_2_R, cannabinoid receptor type 2; FAAH, fatty acid amide hydrolase; i.u., intrauterine; i.v., intravenous; MAGL, monoacylglycerol lipase; PVN, paraventricular nucleus of the hypothalamus; SHR, spontaneously hypertensive rats; WKY, Wistar Kyoto rats.

### 3.4. Influence of Cannabinoids on ACE2 Activity and Other Parameters Relevant in the Fight against COVID-19

The severe acute respiratory syndrome coronavirus type 2 (SARS-CoV-2) infection was rapidly spreading worldwide and led to over a hundred million detected infections and more than 6,200,000 deaths by 26 May 2022 (https://covid19.who.int/ (accessed on 5 May 2022)). Problems with vaccination of part of the population and mutations of the virus mean that SARS-CoV-2 infections may continue for many more years [26]; new promising therapies against COVID-19 are constantly being proposed. Recently, many studies have been published about the relationship between the RAS and the pathophysiology and severity of COVID-19. A key role is played by ACE2, which, however, represents a “double-edged sword” [26]. On the one hand, ACE2 is a SARS-CoV-2 entry receptor, and increased ACE2 expression may facilitate virus entry into cells [1,2,3]. On the other hand, viral binding reduces ACE2 levels at the cell surface, and the shift of the RAS balance towards the deleterious over the protective axis (increase in the ACE/ACE2 ratio) may lead to a progression of disease severity (Figure 3) [95,96]. High ACE/ACE2 ratios occur in males, geriatric, and smoking patients, and many pathologies (especially cardiovascular, pulmonary, and renal diseases and obesity) are recognized as comorbidities that may aggravate the COVID-19 infection. Low ACE/ACE2 ratios are found in women, physically trained individuals, and patients well-treated with ACE inhibitors. The beneficial properties of ACE2 can be achieved either by inhibiting the ACE/Ang II/AT_1_R axis or by promoting the ACE2/Ang 1-7/MasR axis as a result of reducing the ACE/ACE2 ratio and/or by directly stimulating RAS anti-inflammatory components such as soluble ACE2, Ang 1-7 analogs, MasR, or AT_2_R agonists (Figure 3).

Cannabinoids have been proposed as potential therapies or adjuvant drugs against the SARS-CoV-2 infection from the beginning of the COVID-19 pandemic because of their anti-inflammatory and immunomodulatory activities [4,90,97,98].

Two previous publications, which appeared before the onset of the COVID-19 pandemic, showed that modifying the (endo)cannabinoid system may modulate ACE activity/level (Table 2). Thus, chronic administration of the endocannabinoid-like compound PEA reduced ACE levels in the aorta and kidney of SHR, which were enhanced compared to the normotensive WKY [50]. Furthermore, the CB_2_R agonist JWH133 increased the ACE level in N2a cells in a manner sensitive to the CB_2_R antagonist AM630; the ACE level in hippocampus slices from CB_2_^−/−^ mice was lower than in their wild-type littermates [62]. Unfortunately, the promising observations have not been confirmed in conscious (mRen2)27 hypertensive rats (Table 2) and in human iPSC-cardiomyocytes infected with SARS-CoV-2 (Table 4). Thus, although chronic administration of rimonabant reduced BP and HR in hypertensive rats, the level of ACE was not affected [47]. Moreover, although the potent CB_1_/CB_2_R agonist WIN55212-2 inhibited cytotoxicity and the release of proinflammatory cytokines, it failed to affect ACE2 levels [99]. Concerning the latter study (and other studies listed in Table 4), it is of interest that tocilizumab, an antibody acting against the receptor of the proinflammatory cytokine interleukin-6, represents an important therapeutic strategy for the treatment of severe forms of SARS-CoV-2 [100].

**Table 4 ijms-23-06350-t004:** Influence of cannabinoids on ACE2 activity and other relevant effects in the fight against COVID-19.

Model	Agonists	Effects	Final Conclusion of Authors	References
human iPSC- cardiomyocytes infected with SARS-CoV-2	WIN55212-2	↔ ACE2 levels; ↔viral infection and replication; ↓release of proinflammatory cytokines and cytotoxic damage	therapeutic potential of cannabinoids in protecting the heart against SARS-CoV-2 infection is not related to modification of ACE2 levels	[99]
in silico docking studies	CBD THC CBN	CBD: hACE2 (↓), main virus protease activity ↓ THC: hACE2 and main virus protease (↓) CBN: inactive	THC and CBD might inhibit the SARS-CoV-2 infection via their influence on hACE2 and viral proteases	[101]
in silico docking studies	8 phyto-compounds derived from cannabis, including CBD, THC, and CVN	CBD and CVN showed the strongest potency in docking to ACE2, TMPRSS2, NRP1, IL-6, and TNF-α	CBD and CVN may be beneficial for the treatment of COVID-19 and post-COVID-19 neuronal symptoms	[102]
artificial 3D human models of oral, airway, and intestinal tissues treated with TNF-α and IFN-γ	13 high-CBD *Cannabis sativa* extracts	↓ACE2 and TMPRSS2 in oral, lung, and intestinal epithelia constituting important routes of SARS-CoV-2 invasion	*Cannabis sativa* extracts may become a useful and safe addition to the prevention/treatment of COVID-19 as an adjunct therapy; the modulation of ACE2 levels may be an effective strategy for decreasing disease susceptibility	[103]
alveolar epithelial A549 cell line macrophage cell line KG1	extract from *Cannabis sativa* strain Arbel (CBD, CBG, THVC, and terpenes)	**A549**: ↓ACE2 expression together with ↓IL-6, IL-8, CCL2 **macrophage**: ↑IL-6 and ↑IL-8 levels	further studies are needed to determine the therapeutic significance of cannabis in COVID-19 treatment due to its positive (A549 cells) and negative effects (macrophages)	[104]
human colon Caco-2 cell line	CBD	↓ACE2 (concentration-dependent) ↑cell viability, ↓all proinflammatory markers	further studies are needed to clarify the consequences of ACE2 down-regulation and its impact on susceptibility to SARS-CoV-2	[105]
human lung fibroblast WI-38	high-CBD/low-THC cannabis extracts CBD	↓ACE2, TMPRSS, COX2, IL-6, and IL-8 ↓ACE2, TMPRSS	further studies are needed to identify the proper ratios of a combination of single ingredients to find an ideal formulation for future potential clinical studies/use	[31]
human H1299 lung adenocarcinoma cells	industrial hemp **essential oil**: E-caryophyllene and α-pinene were the prominent terpenes and CBDA was the main terpenophenol	↓gene expression of ACE2 and TMPRSS2	hemp essential oils are promising agents to be further investigated with the final goal of optimizing their use in protective devices for counteracting the SARS-CoV-2 virus entry into the human host	[106]
Caco-2 293T-ACE2 Vero E6 cell lines	extracts of hemp and isolates of specific cannabinoids: CBDA and CBGA	cannabinoid acids (CBDA and CBGA) lower SARS-CoV-2 entry into Vero E6 cells through spike binding	CBDA and CBGA (allosterically) block cellular entry of pseudovirus and live SARS-CoV-2 alpha variant B.1.1.7 and beta variant B.1.351	[107]

↓—decrease; (↓)—moderate decrease; ↑—increase; ↔—no change; ACE2, angiotensin-converting enzyme 2; Caco-2, human colorectal adenocarcinoma cell line; CBD, cannabidiol; CBDA, cannabidiolic acid; CBG, cannabigerol; CBGA; cannabigerolic acid; CBN, cannabinol; CCL2, C–C Motif Chemokine Ligands (CCLs) 2; COX2, cyclooxygenase 2; COVID-19; coronavirus disease 2019; CVN, cannabivarin; hACE2, human angiotensin-converting enzyme; hiPSC-CMs, human iPSC-derived cardiomyocytes; IL-6, interleukin-6; IL-8, interleukin-8; NRP1, neuropilin-1; SARS-CoV-2, severe acute respiratory syndrome coronavirus 2; TMPRSS2, transmembrane serine protease 2; THC, Δ⁹-tetrahydrocannabinol; TNF-α, tumor necrosis factor α.

Research dedicated to a potential therapeutic role of cannabinoids in the fight against COVID-19 significantly increased from the beginning of the pandemic onward. As shown in Table 4, the results are related mainly to changes in ACE2 activity/level under treatment of various phytocannabinoids. Thus, in an in silico docking study, both CBD and THC revealed a moderate inhibitory effect against human ACE2 and a strong (CBD) and moderate (THC) inhibitory effect on the main virus protease [101]. In another in silico study, CBD and cannabivarin were the most potent among 8 compounds derived from the cannabis plant to inhibit ACE2 [102]. Moreover, CBD decreased ACE2 expression in human 3D tissue models of oral, airway, and intestinal tissues (that serve as routes for the invasion of SARS-CoV-2) [103] and in human cell lines ([31,105]; Table 4).

Cannabis contains cannabinoids, flavonoids, diterpenes, triterpenes, and lignans [107], which may show a synergistic, so-called “entourage” effect with CBD, i.e., the whole plant extract can have a higher potency and can be more beneficial than the individual compounds [31,108,109]. This putative entourage effect suggested previously for the therapeutic activity of CBD in COVID-19 infection (reviewed in [4]) does not appear to be mediated by direct CB_1_R or CB_2_R activation [110]. As shown in Table 3, high-CBD cannabis extracts downregulated ACE2 expression in 3D models of the oral cavity, intestinal, and lung epithelia [103]. Likewise, high-CBD/low-THC cannabis extracts [31], CBD-enriched Cannabis sativa Arbel strain extract [104], or industrial hemp oil containing terpenes and cannabidiolic acid (CBDA) and cannabigerolic acid (CBGA) [106,107] reduced ACE expression in cell lines [31,104,106]. Moreover, CBD-enriched extracts prevented infection of human epithelial cells by a pseudovirus expressing the SARS-CoV-2 spike protein and the entry of the SARS-CoV-2 alpha variant B.1.1.7 and the beta variant B.1.351 into cells at concentrations that are clinically achievable [107]. 

It should be highlighted that, as shown in Table 4, cannabis extracts, including various phytocannabinoids and terpenes, not only down-regulate ACE2 but can also suppress transmembrane serine protease 2 (TMPRSS2; another virus entry site into the cytoplasm of host cells) [31,103,106], main viral protease [101], neuropilin 1 (NRP1) [102], or inflammation-promoting agents such as IL-6 or IL-8 [31,102,104], CCL2 [104] or cyclooxygenase-2 [31]. However, one should keep in mind that CBD itself is characterized by a bell-shaped dose-response curve associated with a narrow therapeutic window, which makes its effective clinical use difficult. Accordingly, the phytocannabinoid formulations (e.g., CBD with CBG and THCV) may show better and safer activity because such treatment can prevent the bell-shaped dose-response typical for CBD [104].

To summarize the 9 papers in Table 4, cannabinoids in plant extracts may inhibit SARS-CoV-2 via different mechanisms, potentially leading to enhanced effectiveness compared to the individual compounds [4,31,107]. Unfortunately, proinflammatory effects may occur as well [31,103,104]. Thus, their potential therapeutic use in the fight against COVID-19 (e.g., as adjunct therapy) must be approached with caution and requires further research.

### 3.5. Pharmacokinetic Interactions between Drugs Acting on the (Endo)Cannabinoid and Renin-Angiotensin Systems

This review is dedicated to interactions between the (endo)cannabinoid and renin-angiotensin systems at the cellular level. However, if one considers the combined administration of drugs acting on either system, possible pharmacokinetic interactions should also be considered, i.e., one drug may interfere with the metabolism or the transport of another. Unexpectedly, this type of interaction escaped our PubMed-based search. In detail, THC, and, in particular, CBD, is known to interact with drug-metabolizing enzymes like CYP2C8/9 in the liver [111,112,113,114,115]. Moreover, they may affect transport proteins like the bile salt export pump important for losartan and telmisartan excretion. Third, they may, like ACE inhibitors, increase transaminase activity, thereby enhancing their side effects [114]. It is of interest that a wide range of over-the-counter marijuana and CBD-based products is now available, frequently characterized by an unknown CBD content (e.g., [4]). Thus, one should consider the possibility of unexpected pharmacokinetic interactions with RAS-based drugs against hypertension or heart failure.

In summary, strictly speaking, the interactions described in this section do not represent crosstalk between the ECS and the RAS but rather an interplay between individual drugs acting on either system. This interplay can be explained by the fact that the involved drugs, accidentally, share chemical properties, with the consequence that they are common substrates of metabolizing enzymes or transport mechanisms. The practical relevance is high. The concentration or half-time of one drug may be increased by the other, or side effects, e.g., on the liver, may be exaggerated.

## 4. Conclusions

During the past 15 years, there has been a noteworthy progression in the number of publications regarding the cross-talk between the (endo)cannabinoid and renin-angiotensin systems. One important mechanism is that AT_1_R activation by Ang II leads to the release of eCBs (mainly 2-AG), which, by acting at CB_1_Rs, modify the response to AT_1_R stimulation. The interplay, which has been precisely determined in the case of various arteries, should also be examined in detail in other organs, especially in the heart (considered in a single paper only). Another major mechanism is that CB_1_R antagonists diminish AT_1_R levels. A third mechanism is that phytocannabinoids influence ACE2. Moreover, the following interesting observations shown so far only in very few publications require further studies, i.e., (1) the existence of a potential cross-talk between the protective axis of the RAS (Ang 1-7 or AT_2_Rs) with components of the (endo)cannabinoid system, (2) the influence of Ang II on components of the endocannabinoid system like the levels of CB_1_Rs, CB_2_Rs or the AEA transporter and (3) the (patho)physiological significance of AT_1_R-CB_1_R heteromerization. Beyond the specific cross-talk between the RAS and the ECS, pharmacokinetic interactions have to be considered. They occur because single drugs acting on the RAS (telmisartan) or ECS (cannabidiol) accidentally share metabolizing enzymes or transport mechanisms.

The cross-talk may have consequences for drug therapy. (1) Thus, CB_1_R antagonists, the therapeutic significance of which is suggested for various diseases, may enhance (mainly vasoconstriction) or reduce (primarily effects mediated by the CNS) responses elicited by AT_1_Rs. The final reaction depends on whether stimulation of AT_1_Rs and CB_1_Rs induces opposite or the same effects, and potentially beneficial though also detrimental consequences may occur. (2) Despite initially promising results indicating beneficial effects of phytocannabinoids against COVID-19 (especially as adjuvant therapy), additional studies are required to determine their potential therapeutic role and the underlying mechanism (s) in this disease. (3) Pharmacokinetic interactions of drugs acting on the RAS and ECS may lead to an increase in the concentration and half-time of one of the drugs, and to exaggerated side effects, e.g., in the liver.

## Figures and Tables

**Figure 1 ijms-23-06350-f001:**
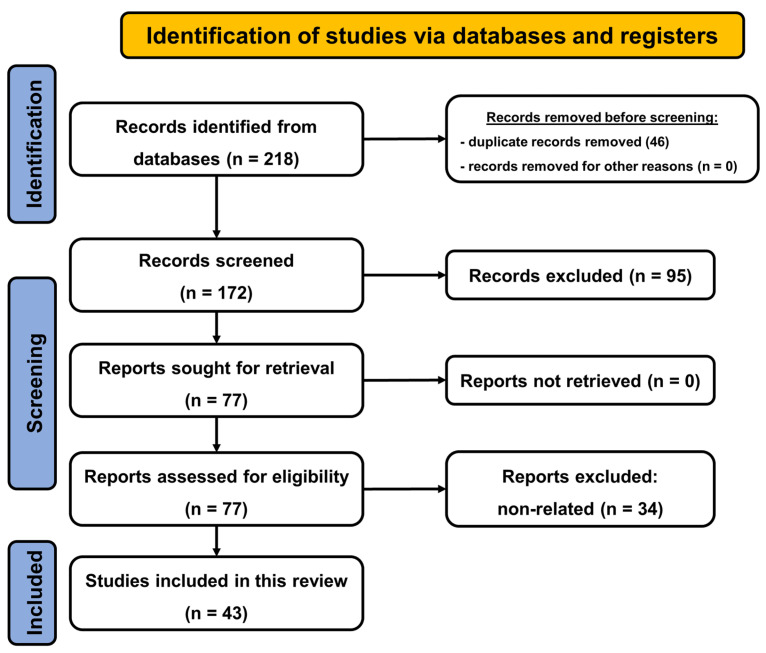
The PRISMA Flow Diagram.

**Figure 2 ijms-23-06350-f002:**
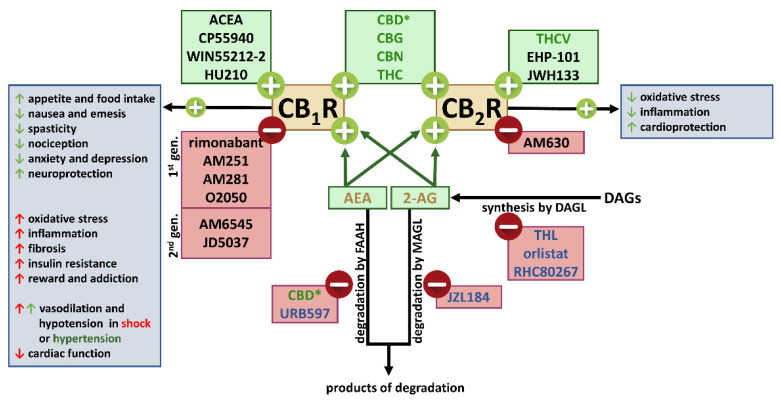
Simplified diagram of compounds modifying the (endo)cannabinoid system as far as they are considered in this review. The ECS comprises endocannabinoids such as anandamide (AEA), 2-arachidonoylglycerol (2-AG), enzymes for their biosynthesis [diacylglycerol lipase (DAGL)] and degradation [fatty acid amide hydrolase (FAAH), and monoacylglycerol lipase (MAGL)] and cannabinoid receptors (CB_1_R, CB_2_R). Green circles with a plus sign describe (partial) agonism at the respective receptor; red circles with a minus sign describe antagonism, inverse agonism, or inhibition at the respective mechanism. Synthetic, plant-derived compounds and endocannabinoids are written in black, green, and brown font, respectively; the blue font is for enzyme inhibitors. Up arrows, increase; down arrows, decrease; green arrows, desired effects; red arrows, undesired effects. 1st gen., first-generation antagonists; 2nd gen., second-generation antagonists; * weak affinity. ACEA, Arachidonyl-2’-chloroethylamide; CBD, cannabidiol; CBG, cannabigerol; CBN, cannabinol; DAGs, diacylglycerols; THC, Δ^9^-tetrahydrocannabinol; THCV, tetrahydrocannabivarin; THL, tetrahydrolipstatin.

**Figure 3 ijms-23-06350-f003:**
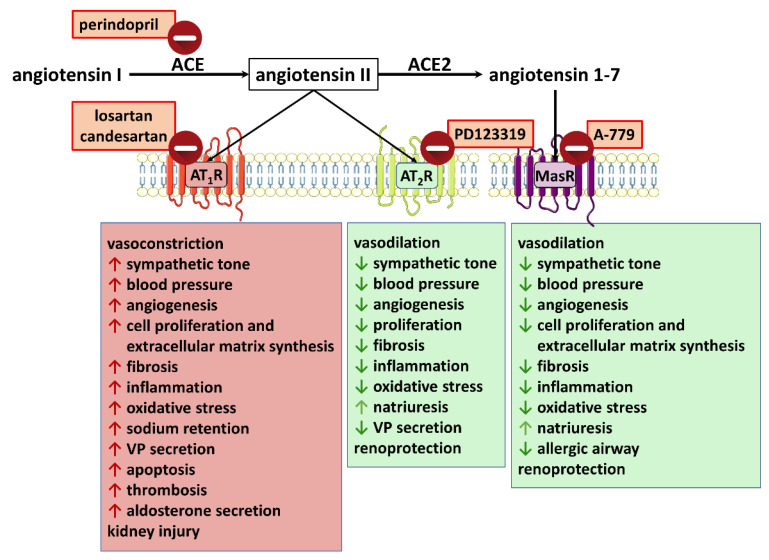
Simplified diagram of the renin-angiotensin system (RAS) and modifying drugs mentioned in this review. The RAS consists of two axes that counteract each other: A deleterious one (so-called classic; red rectangle) containing angiotensin-converting enzyme (ACE)/angiotensin II (Ang II)/Ang II type 1 receptors (AT_1_Rs) and a protective one (so-called alternative; green rectangles) constituted by (i) Ang II type 2 receptors (AT_2_Rs) and (ii) angiotensin-converting enzyme type 2 (ACE2)/angiotensin 1-7) and its Mas receptors (MasR). VP, vasopressin. Red circles with minus signs describe antagonism, inverse agonism, or inhibition at the respective mechanism. Up arrows, increase; down arrows, decrease; green arrows, desired effects; red arrows, undesired effects.

**Figure 4 ijms-23-06350-f004:**
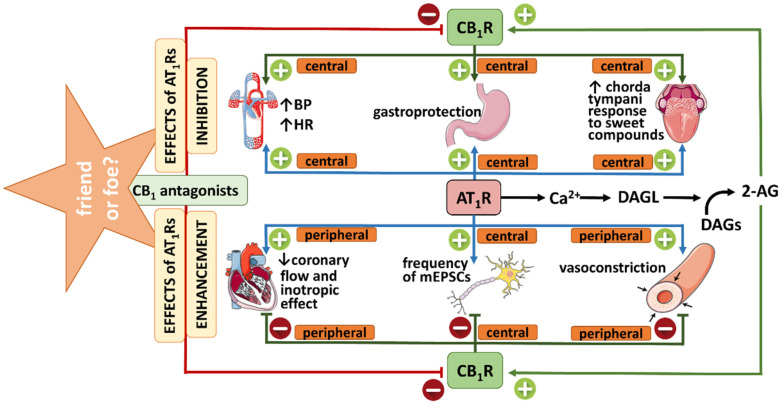
Influence of CB_1_ receptor (CB_1_R) antagonists on the effects elicited by activation of AT_1_ receptors (AT_1_R). Stimulation of AT_1_Rs leads to Ca^2+^-signal generation and rapid biosynthesis of 2-arachidonoylglycerol (2-AG) from diacylglycerols (DAGs) by diacylglycerol lipase (DAGL) activation. If CB_1_R and AT_1_R activation lead to the same effect, CB_1_R antagonism will inhibit the AT_1_R-mediated effect, i.e., will decrease blood pressure (BP) and heart rate (HR), gastro-protection, or the chorda tympani response to sweet compounds (upper part of the figure). In the case of opposite effects of CB_1_R and AT_1_R activation, CB_1_R blockade will enhance the AT_1_R-mediated effects, i.e., its positive inotropic action, its facilitatory effect on miniature excitatory postsynaptic currents (mEPSCs) or its vasoconstrictor effect (lower part of the figure). For the respective literature, see Table 2 and Section 3.2.1.

**Figure 5 ijms-23-06350-f005:**
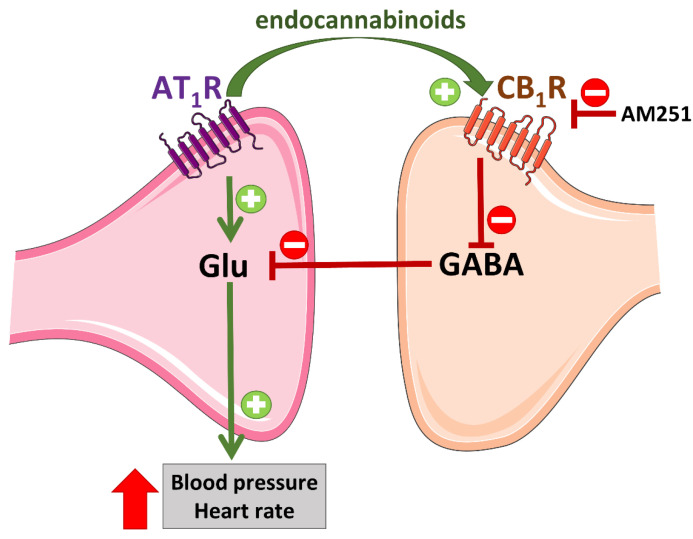
Potential mechanisms of the cross-talk between AT_1_ receptors (AT_1_Rs) and cannabinoid type 1 receptors (CB_1_Rs) in the paraventricular nucleus of the hypothalamus. AT_1_R activation increases blood pressure and heart rate due to a direct and indirectly mediated increase in glutamate (Glu) release. The indirect effect involves an inhibitory γ-aminobutyric acid (GABA) interneuron. In detail, AT_1_R activation increases the release of endocannabinoids (mainly 2-arachidonoyl-glycerol) acting at presynaptic cannabinoid CB_1_ receptors (CB_1_Rs), activation of which decreases the inhibitory influence of GABA on sympathoexcitatory glutamatergic neurons. On the other hand, the CB_1_R antagonist AM251 blocking presynaptic CB_1_Rs increases the inhibitory influence of GABA on glutamatergic neurons excited by AT_1_R activation, resulting in a decreased Ang II-induced pressor response. Facilitatory influences are shown by green arrows and plus signs, whereas inhibitory effects are represented by red bars and minus signs.

## Data Availability

Not applicable.

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
