# Peer review of "Cross-Talk between the (Endo)Cannabinoid and Renin-Angiotensin Systems: Basic Evidence and Potential Therapeutic Significance"

_ijms, 2022, doi:10.3390/ijms23116350_

Round 1

Reviewer 1 Report

ijms-1737289, Potential therapeutic significance of the cross-talk between the (endo)cannabinoid and renin-angiotensin systems

The manuscript presents a solid review of the interactions between the (endo)cannabinoid and renin-angiotensin systems. It is very well written and the presentation is also very good. I did not find any significant mistakes to report, but I find that the article misses a significant section. The title indicates the “Potential therapeutic significance”, but in my opinion this is missing.

The authors should prepare a section in which to present the approved drugs on both signaling systems and how they think the presented data has an impact on the pharmaceutical use of these drugs.

Author Response

Thank you for your compliments and for your comments.  

  • section in which to present the approved drugs on both signaling systems

To this end, a new Table 1 has been added and section 3.1 has been re-phrased accor­dingly.

  • pharmaceutical use of these drugs

To highlight this point, section 3.2.3 has been re-phrased. Moreover, sections 3.3, 3.4, 3.5, 4 and the Summary now end with sentences in which this important topic is addressed.

  • Moreover, we have added the following references:

Page, M.J.; McKenzie, J.E.; Bossuyt, P.M.; Boutron, I.; Hoffmann, T.C.; Mulrow, C.D.; Shamseer, L.; Tetzlaff, J.M.; Akl, E.A.; Brennan, S.E.; et al. The PRISMA 2020 statement: An updated guideline for reporting systematic reviews. BMJ. 2021, 372, 71; DOI: 10.1136/bmj.n71.

www.drugbank.ca, accessed on May 24, 2022

Millar, S.A.; Maguire, R.F.; Yates, A.S.; O'Sullivan, S.E. Towards Better Delivery of Cannabidiol (CBD). Pharmaceuticals (Basel). 2020, 13, 219; DOI: 10.3390/ph13090219

Alsherbiny, M.A.; Li, C.G. Medicinal Cannabis-Potential Drug Interactions. Medicines (Basel). 2018, 6, 3; DOI: 10.3390/medicines6010003.

Doohan, P.T.; Oldfield, L.D.; Arnold, J.C.; Anderson, L.L. Cannabinoid Interactions with Cytochrome P450 Drug Metabolism: a Full-Spectrum Characterization. AAPS J. 2021, 23, 91; DOI: 10.1208/s12248-021-00616-7.

Brown, J.D.; Winterstein, A.G. Potential Adverse Drug Events and Drug-Drug Interactions with Medical and Consumer Cannabidiol (CBD) Use. J. Clin. Med. 2019, 8, 989; DOI: 10.3390/jcm8070989.

Kocis, P.T.; Vrana, K.E. Delta-9-Tetrahydrocannabinol and Cannabidiol Drug-Drug Interactions. Med. Cannabis Cannabinoids. 2020, 3, 61-73; DOI: 10.1159/000507998.

The new part of our manuscript is marked in yellow color.

We believe that we have addressed all the comments of the reviewer. We hope that you now will find our revised manuscript suitable for publication in the Special Issue of International Journal of Molecular Sciences“ entitled “New Insight into Cannabinoid Effects”.

Reviewer 2 Report

This a conventional review of the complex interactions between the endocannabinoid and renin-angiotensin systems. The review has an ambitious goal that, however, would need a well-organized (systematic) scientific approach and clear presentation of the results. The authors need to comply with the current standards of preparing the review and they need to clearly formulate the "question" and use systematic and fully reproducible methods to identify and select (please describe the literature selection process step-by-step, best in a graphic form). The information collected by the authors should be also systematically analyzed and presented. Otherwise this review does not comply to the standard of the reputable journal. 

Author Response

This a conventional review of the complex interactions between the endocannabinoid and re­nin-angiotensin systems. The review has an ambitious goal that, however, would need a well-orga­nized (systematic) scientific approach and clear presentation of the results. The authors need to comply with the current standards of preparing the review and they need to clear­ly for­mulate the "question" and use systematic and fully reproducible methods to identify and se­lect (please describe the literature selection process step-by-step, best in a graphic form). The information collected by the authors should be also systematically analyzed and pre­­sen­ted. Otherwise this review does not comply to the standard of the reputable journal.

  • need to clearly formulate the "question"

We have formulated a primary and secondary aim in section 1. These two aims are re-addressed in section 3.2.3 and in the summarizing parts of 3.3, 3.4, 3.5 and 4. The title of the paper has also been changed.

  • describe the literature selection process step-by-step, best in a graphic form

Section 2 and Figure 1 have been added for this purpose. According to your suggestion we have reorganized our publication and we have added the new figure with the PRISMA Flow Diagram and additional reference: Page, M.J.; McKenzie, J.E.; Bossuyt, P.M.; Boutron, I.; Hoffmann, T.C.; Mulrow, C.D.; Shamseer, L.; Tetzlaff, J.M.; Akl, E.A.; Brennan, S.E.; et al. The PRISMA 2020 statement: An updated guideline for reporting systematic reviews. BMJ. 2021, 372, 71; DOI: 10.1136/bmj.n71.

 information collected by the authors should be also systematically analyzed and presen­ted

Section 3.2.3 has been re-phrased and summarizing sentences have been added to sections 3.3, 3.4, 3.5 and 4.

  • Section 3.5, which is dedicated to pharmacokinetic interactions of single drugs targe­ting the ECS and RAS, has been added. Instead, one paragraph, in which the very likely interaction between presynaptic AT1 and CB1 receptors had been proposed but not shown, has been deleted (the two references had not been entered into Table 2).

The new part of our manuscript is marked in yellow color.

We believe that we have addressed all the comments of the reviewer. We hope that you now will find our revised manuscript suitable for publication in the Special Issue of International Journal of Molecular Sciences“ entitled “New Insight into Cannabinoid Effects”.

Round 2

Reviewer 2 Report

The authors have addressed all my criticisms and have significantly improved the manuscript.